# Two-year persistence of MERS-CoV-specific antibody and T cell responses after MVA-MERS-S vaccination in healthy adults

Leonie Mayer [1,2,3] ✉, Anahita Fathi[1,2,3,4], Hanna-Marie Weichel [1,2,3], Matthijs P. Raadsen [5], Christine Dahlke[1,2,3], Anna Mykytyn [5], Jordi Rodon[6], Gesche K. Gerresheim[7,8], Merel R. te Marvelde [5], Leonie M. Weskamm [1,2,3], Ilka Grewe[1,2,3], Claudia Schlesner[1,2,3], Marc Lütgehetmann [3,9], Christian Drosten [6,10], Stephan Becker[7,8], Bart L. Haagmans [5], Svenja Hardtke [1,2,3] & Marylyn M. Addo [1,2,3] ✉

MVA-MERS-S, a vaccine candidate against Middle East respiratory syndrome (MERS), was recently evaluated in a randomized, placebo-controlled, double-blind phase 1b clinical trial to assess its safety, immunogenicity, and optimal dosing in healthy adults in Hamburg and Rotterdam. A three-dose regimen was safe and elicited robust spike-specific antibody responses. We extended this trial to assess the two-year durability of MERS-CoV-specific antibody and T cell responses in 48 study participants of the Hamburg cohort. Our findings show that immune responses remain detectable for at least 24 months after the third vaccination. Antibodies persisted at levels comparable to the peak response observed after the second vaccination and were able to cross-neutralize MERS-CoV spike mutants. Although the immune correlates of protection against MERS remain unknown, the observed durability of humoral and cellular immune responses supports the potential of MVA-MERS-S as a promising MERS vaccine candidate and highlights the importance of a booster dose in sustaining long-term immunity.

The Middle East respiratory syndrome coronavirus (MERS-CoV) emerged as a human pathogen in 2012 and has a reported case fatality rate of up to 36%[1,2]. MERS-CoV continues to circulate in camels and infections in humans are reported sporadically. It poses a potential pandemic threat and is considered a priority pathogen for the development of vaccines and therapeutics[3]. Currently, no licensed vaccine is available; however, three vaccine candidates based on the MERS-CoV

spike protein have been shown to be safe and immunogenic in human phase 1 clinical trials: MVA-MERS-S[4–6], ChAdOx1 MERS[7,8], and GLS-5300 DNA MERS-CoV[9].

We recently conducted a randomized, double-blind, placebo-controlled phase 1 clinical trial of MVA-MERS-S, a MERS vaccine candidate based on the replication-deficient Modified Vaccinia virus Ankara (MVA) viral vector. The trial aimed to determine the optimal

[1]Institute for Infection Research and Vaccine Development (IIRVD), University Medical Center Hamburg-Eppendorf, Hamburg, Germany. [2]Department for Clinical Immunology of Infectious Diseases, Bernhard Nocht Institute for Tropical Medicine, Hamburg, Germany. [3]German Center for Infection Research, partner site Hamburg-Lübeck-Borstel-Riems, Hamburg, Germany. [4]First Department of Medicine, Division of Infectious Diseases, University Medical Center Hamburg-Eppendorf, Hamburg, Germany. [5]Department of Viroscience, Erasmus Medical Center, Rotterdam, The Netherlands. [6]Institute of Virology, Charité - Universitätsmedizin Berlin, corporate member of Freie Universität Berlin, Humboldt-Universität zu Berlin and Berlin Institute of Health, Berlin, Germany. [7]Institute of Virology, Philipps University Marburg, Marburg, Germany. [8]German Center for Infection Research, partner site Gießen-Marburg-Langen, Marburg, Germany. [9]Institute of Medical Microbiology, Virology and Hygiene, University Medical Center Hamburg-Eppendorf, Hamburg, Germany. [10]German Center for Infection Research, associated partner Charité, Berlin, Germany. ✉e-mail: l.mayer@uke.de; m.addo@uke.de

dose, prime-boost interval, and impact of a booster dose in 139 healthy adults (NCT04119440). The primary and secondary outcomes of the trial have been published by Raadsen et al.[10]. Prime-boost vaccination with MVA-MERS-S elicited robust binding and neutralizing antibody responses, with higher titers observed when the prime-boost interval for the high dose group ($10^8$ plaque-forming units [PFU]) was extended from 28 to 56 days. Notably, administering a third dose six months later significantly enhanced the antibody response, leading to peak titers that were comparable across the different vaccine doses and prime-boost intervals, but overall higher than those observed after the second dose[10].

While the immunological mechanisms mediating protection against MERS remain mostly unknown, vaccine efficacy studies of the closely related betacoronavirus SARS-CoV-2 have demonstrated that antibody responses serve as the best immunological correlate of protection across different vaccine platforms[11,12]. However, unlike live-attenuated vaccines, which can induce lifelong immunity[13], the antibody responses elicited by most subunit, mRNA, and non-replicating viral vector vaccines licensed to date tend to wane over time[14]. Assessing the relationship between declining antibody titers and duration of protection against COVID-19 has been challenging due to limited long-term follow-up studies and the occurrence of breakthrough infections with SARS-CoV-2 variants[15–18].

To evaluate the durability of vaccine-induced immunity against MERS-CoV and the impact of different dosing regimens, we extended the MVA-MERS-S phase 1b clinical trial to allow for a 24-month follow-up of 48 study participants. Here, we show that binding and neutralizing antibodies, as well as T cell responses are maintained for at least 24 months following the third MVA-MERS-S vaccination. While we previously showed that MVA-MERS-S vaccination does not elicit cross-reactive binding antibodies against SARS-CoV-2[19], we also assessed SARS-CoV-2 antibody responses and recorded SARS-CoV-2 infections and vaccinations in this study to determine whether they influence the persistence of MERS-CoV-specific antibody responses.

## Results

### Demographics and safety

We originally enrolled 139 participants in the phase 1b clinical trial between 2021 and 2022 at the study sites in Hamburg and Rotterdam. According to the trial design, participants had been randomized by vaccine dose and V1-V2 time interval to one of four treatment groups: the 28-day $10^7$ PFU group, the 28-day $10^8$ PFU group, the 56-day $10^7$ PFU group, the 56-day $10^8$ PFU group, or to the placebo group (Fig. 1).

Of the 74 participants enrolled in Hamburg, 55 re-consented to participate in the subsequent 24-month follow-up study (Fig. 2). All 55 participants received at least one injection and were included in the safety analysis. Seven participants of the treatment groups had missed one or more MVA-MERS-S vaccinations. The remaining 48 participants had received all three MVA-MERS-S vaccinations (treatment groups) or placebo doses (placebo group), as defined by the modified intention-to-treat (mITT) criteria of the study[10], and were included in the longitudinal immunogenicity analysis. Four participants were lost to follow-up after V3M6 or V3M12.

Table 1 summarizes the demographics of the mITT immunogenicity set at the time of screening, stratified by study group. The demographics of the safety set are shown in Supplementary Table 1. During the follow-up period, nine participants reported to have received COVID-19 vaccinations, four participants received other vaccines within four weeks before a visit, and two participants received immunosuppressive therapy (Supplementary Table 2). No participant received an MVA or Mpox vaccine during the study. A total of 32 cases of COVID-19 or any febrile illness were reported (Supplementary

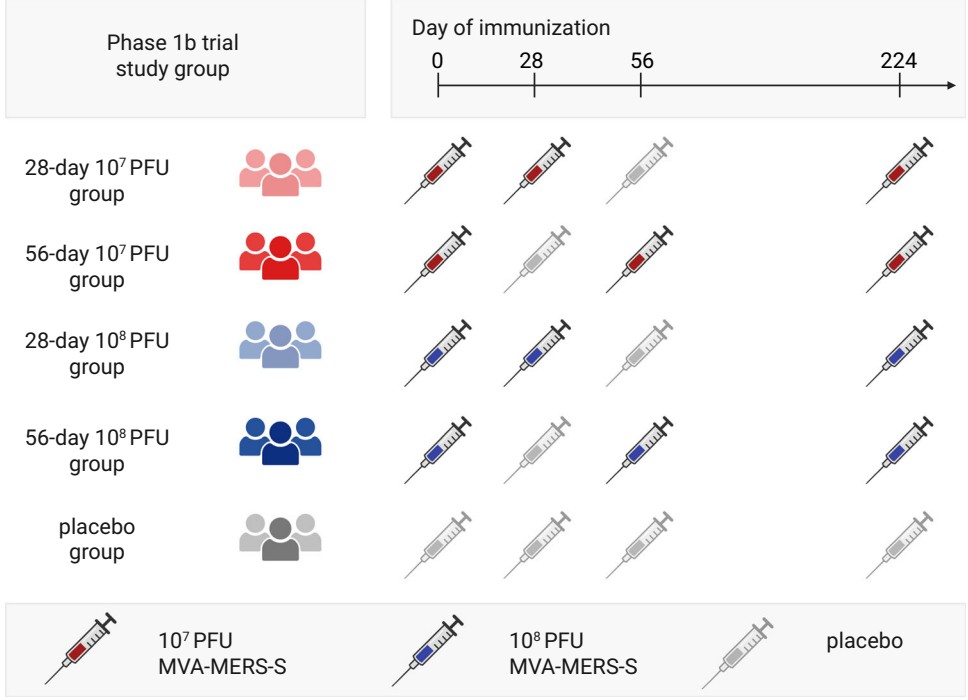

**Fig. 1 | Study design.** Schematic of the study design of the MVA-MERS-S phase 1b clinical trial. Participants were randomized to receive three vaccinations (V1, V2, V3) of MVA-MERS-S of either $10^7$ PFU with a 28-day V1-V2 interval (light red), $10^7$ PFU with a 56-day V1-V2 interval (dark red), $10^8$ PFU with a 28-day V1-V2 interval (light blue) or $10^8$ PFU with a 56-day V1-V2 interval (dark blue). Participants received a placebo dose at day 56 (28-day interval groups) or day 28 (56-day interval groups) to maintain study blinding. V3 was administered at day 224. Participants of the placebo group received four placebo vaccinations (grey). PFU plaque-forming units. Created in BioRender. Mayer, L. (2026) https://BioRender.com/6etvfmw.

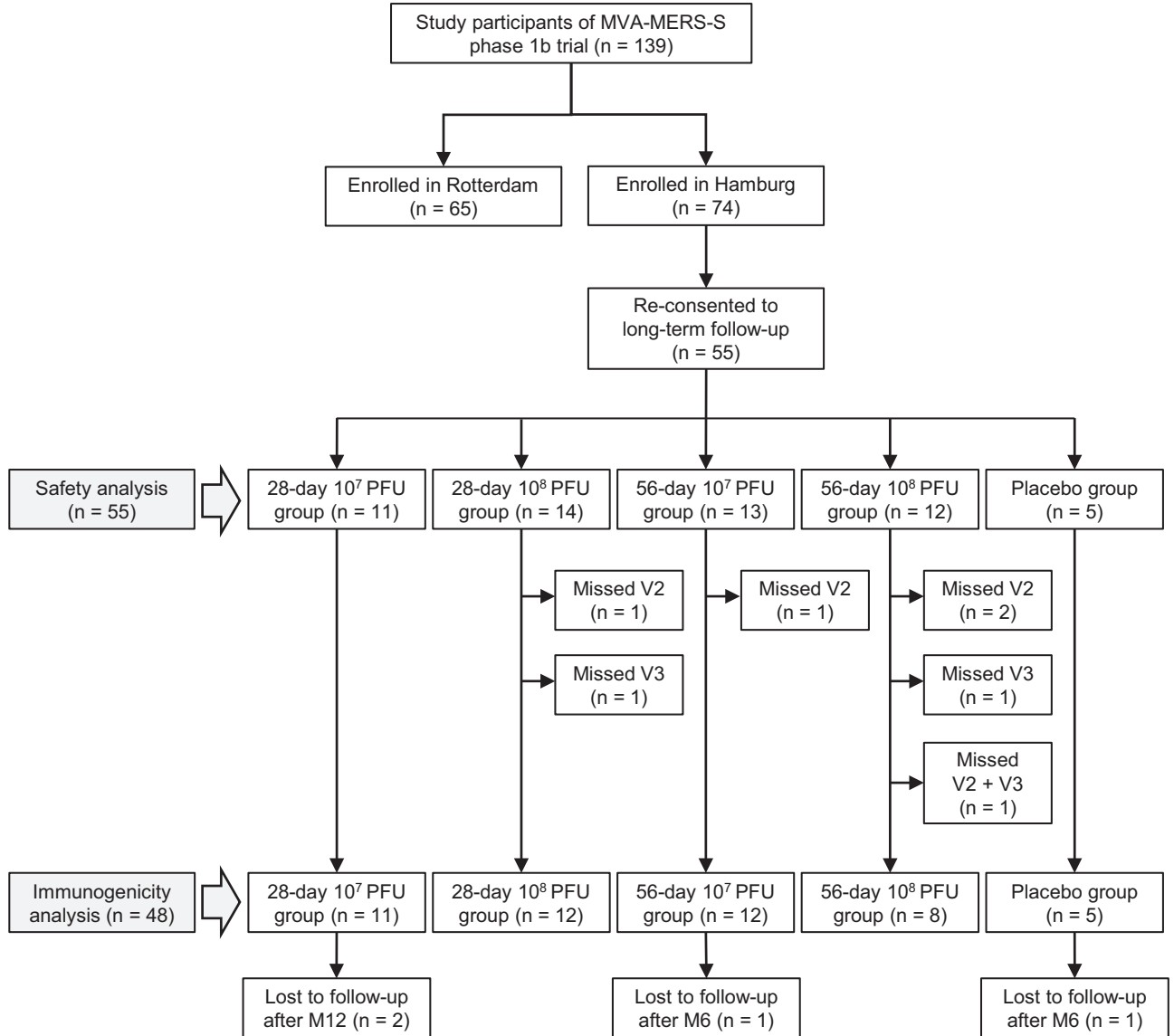

**Fig. 2 | Trial profile.** All study participants enrolled at the Hamburg site were eligible for participation in the extension of the phase 1b trial. Of those, $n = 55$ re-consented and were included in the long-term follow-up. Seven participants missed ≥1 MVA-MERS-S vaccinations and were excluded from the immunogenicity analysis, resulting in a modified intention-to-treat cohort of $n = 48$. Four participants were lost to follow-up and a total of $n = 44$ participants completed the 24-month follow-up. V vaccination, M month, PFU plaque-forming units.

### Table 1 | Demographics of the immunogenicity set

| | 28-day 10⁷ PFU ($n = 11$) | 56-day 10⁷ PFU ($n = 12$) | 28-day 10⁸ PFU ($n = 12$) | 56-day 10⁸ PFU ($n = 8$) | Placebo ($n = 5$) | Total ($n = 48$) |
|---|---|---|---|---|---|---|
| Sex | | | | | | |
| Male – $n$ (%) | 7 (64%) | 6 (50%) | 6 (50%) | 5 (63%) | 1 (20%) | 25 (52%) |
| Female – $n$ (%) | 4 (36%) | 6 (50%) | 6 (50%) | 3 (38%) | 4 (80%) | 23 (48%) |
| Age – years | | | | | | |
| mean (SD) | 30 (±12.7) | 37 (±10.9) | 34 (±11.5) | 35 (±12.1) | 41 (±8.1) | 35 (±11.5) |
| Body-mass index | | | | | | |
| mean (SD) | 24 (±3.5) | 24 (±2.6) | 23 (±2.7) | 24 (±2.2) | 22 (±1.8) | 23 (±2.6) |

Summary of the demographic characteristics of participants of the long-term follow-up study that met the modified intention-to-treat definition, stratified by study group. Data were collected at the screening visit before V1.

Table 2). Six moderate to severe serious adverse events were reported in 4/55 (7%) participants (Supplementary Table 3). All events were considered to be unrelated to the study vaccination. Two participants had short-term stays in countries where MERS-CoV is endemic (Supplementary Table 4).

### Persistence of the MERS-CoV-specific antibody response

To assess the long-term persistence of vaccine-induced MERS-CoV-specific antibody responses following three doses of MVA-MERS-S, we measured MERS-CoV S1-specific IgG (S1 IgG) and full-spike-specific IgG (full S IgG), as well as neutralizing antibody responses at the V3M6,

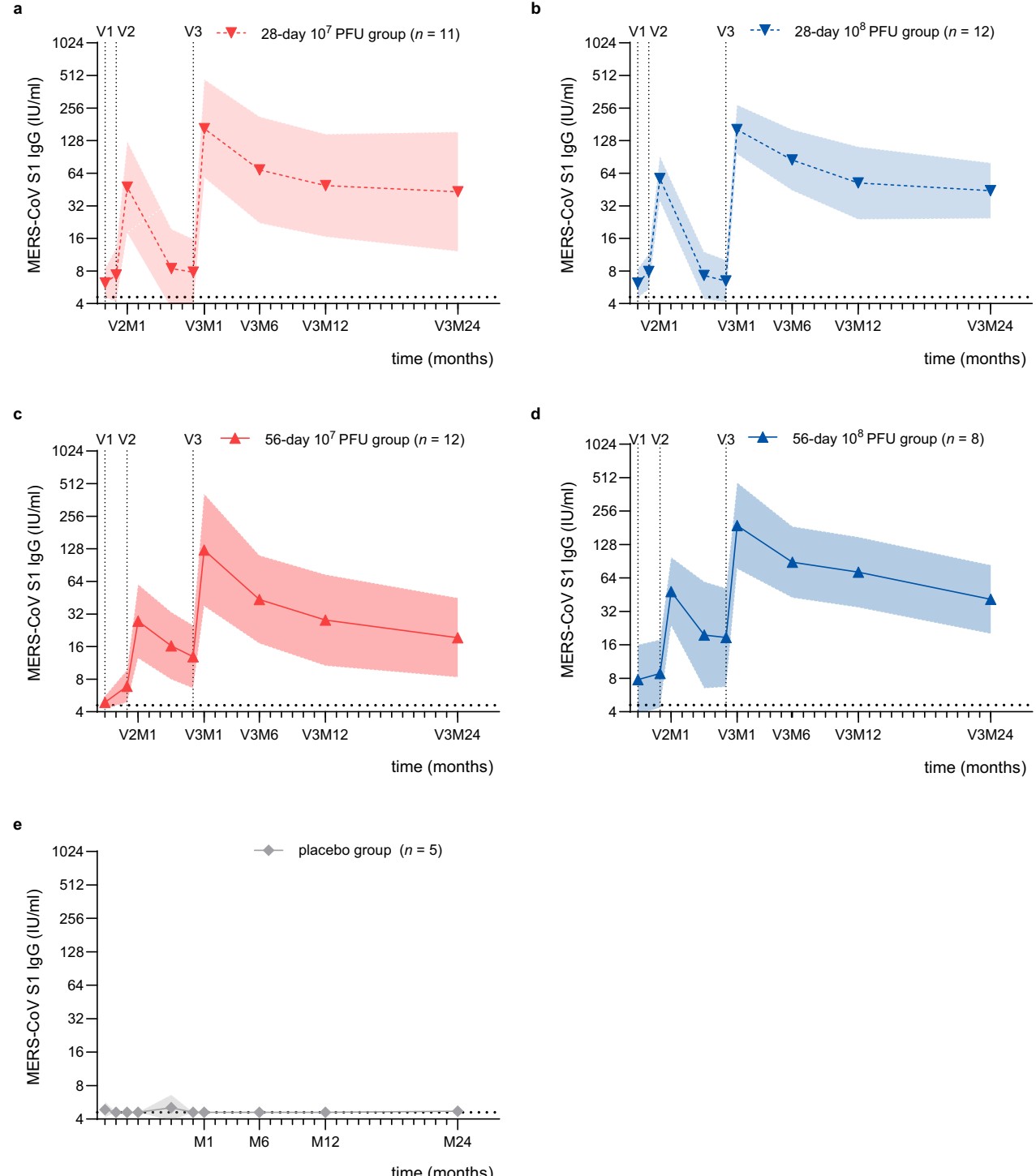

**Fig. 3 | Longitudinal S1 IgG responses elicited by MVA-MERS-S vaccination.**
Longitudinal S1 IgG responses are shown as geometric mean titers with 95% confidence interval bands for the 28-day $10^7$ PFU group (**a**), 28-day $10^8$ PFU group (**b**), 56-day $10^7$ PFU group (**c**), 56-day $10^8$ PFU group (**d**), and placebo group (**e**). S1 IgG titers were calibrated to the WHO standard (IU/ml, y-axis) and shown as a function of time since last vaccination (months, x-axis). The dotted horizontal lines indicate the lower limit of detection (4.6 IU/ml). The dotted vertical lines show the timepoints of vaccinations V1, V2 and V3. V vaccination, M month, S spike, PFU plaque-forming units, IU international units. Source data are provided as a Source Data file.

V3M12, and V3M24 timepoints. Figure 3 shows the longitudinal S1 IgG response of the immunogenicity set from baseline (pre-V1) through the last timepoint of the 24-month follow-up (V3M24), stratified by study group: 28-day $10^7$ PFU (a), 28-day $10^8$ PFU (b), 56-day $10^7$ PFU (c), and 56-day $10^8$ PFU (d), placebo (e). Geometric mean titers (GMTs) of S1 IgG at the timepoints shown in Fig. 3 are summarized in

Supplementary Table 5. Robust S1 IgG responses were induced after the second dose (V2M1), were further boosted by V3 and peaked at V3M1 with comparable titers across all treatment groups[10]. Titers then gradually declined but remained detectable through V3M24 (Fig. 3). In total, 60% (24/40) of vaccinated participants remained S1 IgG seropositive at V3M24, compared to 86% at V3M1 (Supplementary

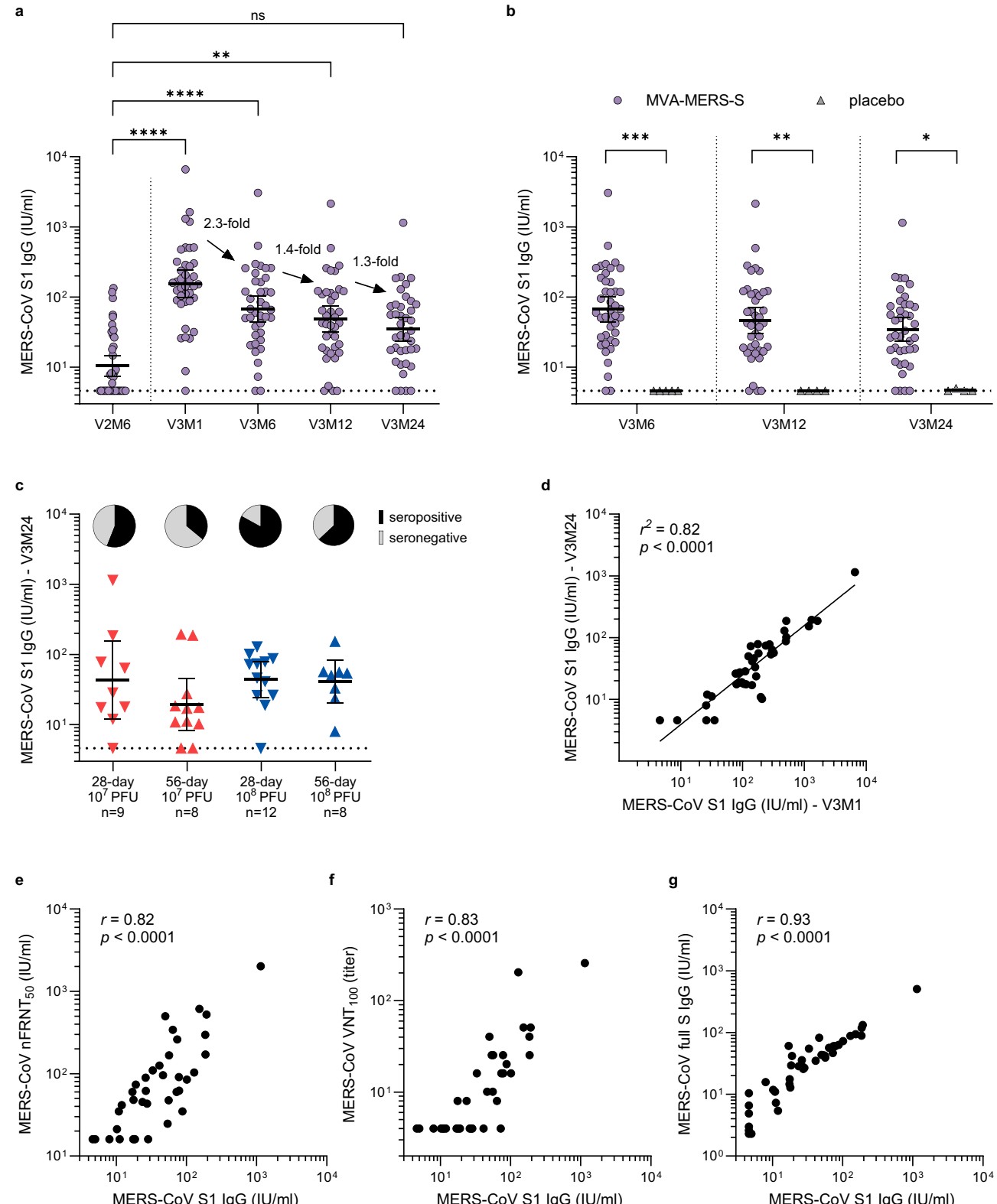

Table 6). Similar dynamics were observed for MERS-CoV-neutralizing antibody responses, measured by pseudovirus (Supplementary Fig. 1) and live-virus neutralization assays (Supplementary Fig. 2). At V3M24, 75% (30/40) and 50% (20/40) of participants maintained detectable pseudovirus and live-virus neutralizing titers, respectively. The S1 IgG and neutralizing responses of individual participants are shown in Supplementary Fig. 3.

As shown in Fig. 4a, peak S1 IgG titers in vaccinated individuals at V3M1 decreased 2.3-fold until 6 months after vaccination (V3M6) and then waned more gradually, with a 1.4-fold decrease by month 12 (V3M12) and a further 1.3-fold decrease by month 24 (V3M24). Notably, antibody responses remained at higher titers after V3 compared to antibody levels achieved after two immunizations (V2). Compared to the S1 IgG GMT at V2M6 (10 IU/ml, 95% CI: 7.7–14.8) the S1 IgG GMTs

**Fig. 4 | Persistence of vaccine-induced antibody responses. a** S1 IgG titers of all MVA-MERS-S-vaccinated participants ($n = 39$) after V2 at month 6 (V2M6) compared to after V3 at months 1 (V3M1, $p < 0.0001$), month 6 (V3M6, $p < 0.0001$), month 12 (V3M12, $p = 0.0031$) and month 24 (V3M24, $p > 0.99$), showing the fold-reduction of geometric mean titers after V3. **b** S1 IgG titers in vaccinated individuals (purple, $n = 39$) compared to the placebo group (grey, $n = 4$) at V3M6 ($p = 0.0002$), V3M12 ($p = 0.0076$), V3M24 ($p = 0.038$). **c** Comparison of V3M24 titers in the different treatment groups. Pie charts show proportions of seropositive (black) and seronegative (grey) individuals defined as having at least 4-fold higher titers at V3M24 compared to baseline before first vaccination. Data are shown as individual points and geometric mean titers with 95% confidence interval. The dotted horizontal lines indicate the lower limit of detection (4.6 IU/ml) (**a**–**c**). **d** Simple linear regression of S1 IgG titers in vaccinated individuals at V3M1 and V3M24 ($n = 40$, $r^2 = 0.82$, $p < 0.0001$). **e** Spearman correlation of V3M24 S1 IgG responses of all participants ($n = 44$) with V3M24 (**e**) neutralizing responses measured by pseudovirus neutralization assay ($r = 0.82$, $p < 0.0001$) and (**f**) live-virus neutralization assay ($r = 0.83$, $p < 0.0001$) as well as (**g**) full S IgG responses ($r = 0.93$, $p < 0.0001$). S1 IgG titers were compared using Friedman test (**a**) or Kruskal–Wallis test (**b**, **c**) and adjusted using Dunn's multiple comparison. Associations were investigated using simple linear regression (**d**) and Spearman's correlation (**e**–**g**). Statistical tests are two-sided. V vaccination, M month, S spike, IU international units, nFRNT$_{50}$ normalized pseudovirus 50% focus reduction neutralization assay, VNT$_{100}$ live-virus neutralization test, $*p < 0.05$, $**p < 0.01$, $***p < 0.001$, $****p < 0.0001$. Source data are provided as a Source Data file.

were significantly higher at V3M6 (68 IU/ml, 95% CI: 44.2–103.9, $p < 0.0001$) and V3M12 (49 IU/ml, 95% CI: 31.4–75.1, $p < 0.0001$), as well as at V3M24 (35 IU/ml, 95% CI: 23.4–51.8, $p = 0.4436$) although not reaching significance (Fig. 4a). Antibody titers remained higher in vaccinated individuals compared to the placebo group at all long-term follow-up timepoints (V3M6 ($p = 0.0002$), V3M12 ($p = 0.0076$), and V3M24 ($p = 0.0376$) (Fig. 4b)). No statistically significant differences in titers at V3M24 were observed between the treatment groups (Fig. 4c). At V3M24, 56%, 36%, 83%, and 63% of participants in the 28-day $10^7$ PFU, 56-day $10^7$ PFU, 28-day $10^8$ PFU, and 56-day $10^8$ PFU groups, respectively, remained seropositive (Fig. 4c, Supplementary Table 6). Participants in the placebo group remained seronegative at all follow-up timepoints.

We observed a significant positive correlation between peak antibody titers (V3M1) and long-term antibody titers two years following the last vaccination (V3M24; $r = 0.89$, $p < 0.0001$, $r^2 = 0.82$, $p < 0.0001$, Fig. 4d). The GMT of high responders (defined as participants with S1 IgG titers above the mean at V3M1) decreased by 5.5-fold to 76 IU/ml at V3M24, whereas the GMT of the low responders (defined as participants with S1 IgG titers below the mean at V3M1) decreased by 3.5-fold to 17 IU/ml at V3M24. The longitudinal S1 IgG titers of high and low responders are shown in Supplementary Fig. 4. A demographic comparison of high and low responders is provided in Supplementary Table 7. S1 IgG responses at V3M24 showed a strong positive correlation with MERS-CoV-neutralizing antibody responses measured by pseudovirus assay ($r = 0.82$, 95% CI: 0.69–0.90, $p < 0.0001$, Fig. 3e) and live-virus assay ($r = 0.83$, 95% CI: 0.71–0.91, $p < 0.0001$, Fig. 3f), as well as full S IgG ($r = 0.92$, 95% CI: 0.87–0.96, $p < 0.0001$, Fig. 3g).

### Impact of SARS-CoV-2 exposure and anti-vector immunity
In the immunogenicity set, 47/48 (98%) participants were SARS-CoV-2 seropositive at baseline before MVA-MERS-S vaccination (Supplementary Fig. 5a) and 42/48 (88%) showed an increase in SARS-CoV-2 nucleocapsid antibodies indicative of a SARS-CoV-2 infection during the 24-month follow-up (Supplementary Fig. 5b). There was no significant correlation between pre-existing SARS-CoV-2 S IgG titers and MERS-CoV S1 IgG at V3M1 ($r = -0.017$, 95% CI: $-0.31$–0.28, $p = 0.91$, Supplementary Fig. 5c) and V3M24 ($r = 0.14$, 95% CI: $-0.17$–0.43, $p = 0.37$, Supplementary Fig. 5d), as well as MERS-CoV neutralizing antibody titers at V3M1 ($r = 0.020$, 95% CI: $-0.27$–0.31, $p = 0.90$, Supplementary Fig. 5e) and V3M24 ($r = 0.11$, 95% CI: $-0.20$–0.40, $p = 0.48$, Supplementary Fig. 5f).

We observed no significant correlation between pre-existing neutralizing antibody titers against the MVA vector and peak S1 IgG titers at V3M1 ($r = -0.17$, 95% CI: $-0.44$–0.13, $p = 0.25$, Supplementary Fig 6a) or persistent S1 IgG titers at V3M24 ($r = -0.16$, 95% CI: $-0.45$–0.15, $p = 0.29$ Supplementary Fig. 6b). MVA-neutralizing titers elicited by the first two vaccinations did not correlate with peak S1 IgG titers at V3M1 ($r = 0.19$, 95% CI: $-0.11$–0.46, $p = 0.20$, Supplementary Fig. 6c) or persistent S1 IgG titers at V3M24 ($r = 0.13$, 95% CI: $-0.073$–0.50, $p = 0.38$, Supplementary Fig. 6d).

### Cross-neutralization of MERS-CoV spike variants
To assess whether MVA-MERS-S elicits cross-neutralization against clinically relevant MERS-CoV variants, we generated VSVpp harboring spike mutations D510G or I529T (Fig. 5a), which emerged in the 2015 outbreak in South Korea, and measured virus neutralization in ten serum samples. As shown in Fig. 5b, GMTs against wild-type, mutant D510G and mutant I529T spike were 830 (95% CI: 298–2312), 1091 (95% CI: 355–3355), and 1074 (95% CI: 381–3028), respectively. Compared to the wild-type, there was no significant reduction in neutralization of mutant D510G (Fig. 5c) or mutant I529T (Fig. 5d), indicating that MVA-MERS-S elicits antibodies that can cross-neutralize these variants.

### Persistence of the MERS-CoV-specific T cell response
Next, we analyzed if vaccine-induced T cell responses persist until two years after vaccination by restimulating fresh whole blood samples at the V3M24 timepoint with an overlapping spike peptide pool and measuring cytokine release by ELISA (Fig. 6a). A shown in Fig. 6b, compared to the median IFN-γ response of the placebo group (6.3 pg/ml, 1.1–10.4), the median IFN-γ response of the MVA-MERS-S-vaccinated group was significantly higher (31.5 pg/ml, 95% CI: 13.5–55.2, $p = 0.0106$). Similarly, compared to the median IL-2 levels of the placebo group (4.3 pg/ml, 95% CI: 2.2–10.0), the median IL-2 response of the MVA-MERS-S-vaccinated group was significantly higher (35.1 pg/ml, 95% CI: 24.3–49.0, $p = 0.0023$, Fig. 6c). There was a significant positive correlation between the IFN-γ and IL-2 response ($r = 0.87$, 95% CI: 0.77–0.93, $p < 0.0001$, Fig. 6c).

## Discussion
The induction of durable, protective immunity remains a major challenge in vaccine development. Although three MERS vaccine candidates have undergone phase 1 clinical trials[4,7–9], the durability of immunogenicity remains unknown. In this study, we demonstrated that three-dose vaccination with the MVA-MERS-S candidate vaccine elicits robust immune responses that persist for at least two years, with 75% and 50% of participants maintaining detectable neutralizing antibodies measured by pseudovirus and live-virus assays, respectively, and 60% of participants remaining S1 IgG seropositive. Furthermore, we could show that vaccine-induced T cells secreting IFN-γ and IL-2, indicative of a Th1-biased response, were detectable for at least two years. Two other MERS vaccine candidates, ChAdOx1 MERS and GLS-5300 DNA, have demonstrated antigen-specific immunogenicity in humans, but only limited data on long-term immunity are available. Folegatti et al. reported that one year after single-dose ChadOx1 MERS vaccination, 68% of participants maintained full-spike-specific antibody titers above the assay cut-off and the T cell responses remained significantly above baseline[8]. Booster doses of ChAdOx1 MERS have not been tested in clinical trials; however, in a preclinical model, responses elicited by ChadOx1 MERS could be significantly boosted by subsequent vaccination with an MVA-based vaccine candidate[20]. Modjarrad et al. reported that one year after three-dose GLS-5300 DNA vaccination, 3% of participants had detectable live-virus neutralizing

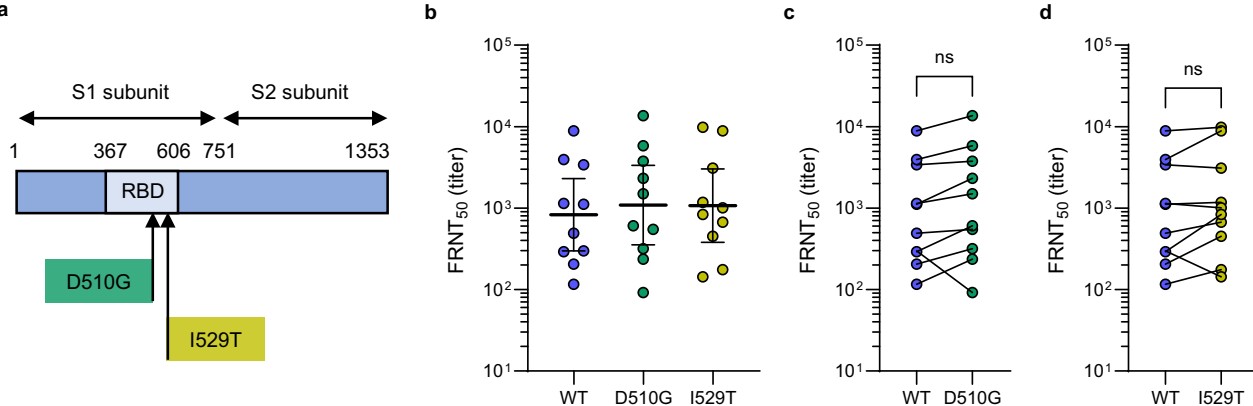

**Fig. 5 | Neutralization of MERS-CoV variants. a** Schematic of the MERS-CoV spike protein showing the location of the D510G and I529T polymorphisms within the receptor-binding domain (RBD) of the S1 subunit. **b** Neutralizing antibody titers in participant sera ($n = 10$, $n = 4$ female, $n = 6$ male) against Vesicular stomatitis virus-based pseudotypes (VSVpp) harboring the wild-type (WT) MERS-CoV spike (blue) or MERS-CoV spike variants D510G (green) and I529T (yellow). Data are shown as individual points (mean of duplicates) and geometric mean titers with 95%

confidence interval. Comparison of neutralization in individual samples ($n = 10$) against wild-type and variant **c** D510G ($p = 0.076$) and **d** I529T ($p = 0.22$). Neutralizing titers were compared using Friedman test and adjusted using Dunn's multiple comparison. Statistical tests are two-sided. S spike, $FRNT_{50}$ pseudovirus 50% focus reduction neutralization assay, aa amino acid, ns not significant. Source data are provided as a Source Data file.

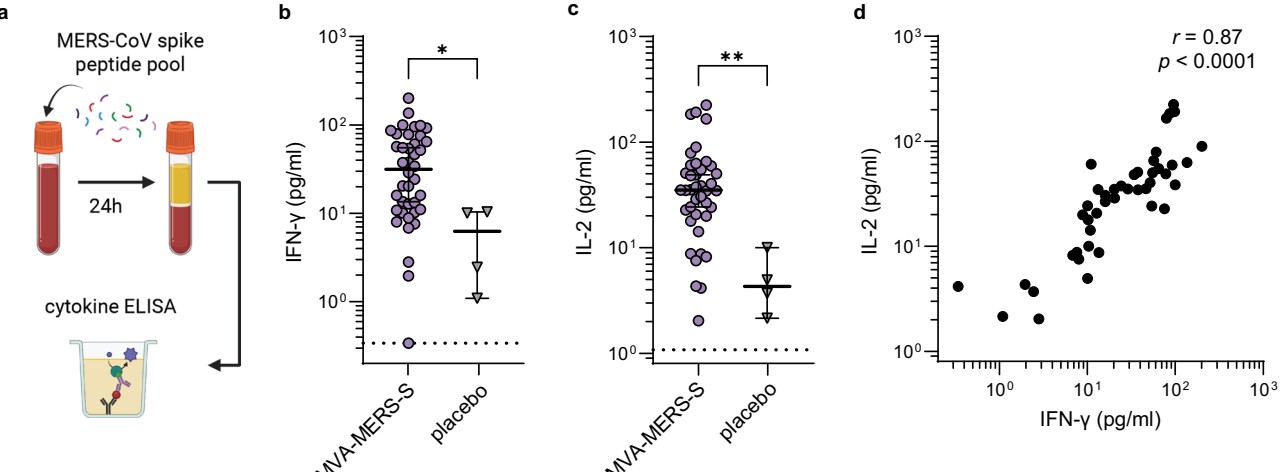

**Fig. 6 | Persistence of MERS-CoV-specific T cell response. a** The T cell responses was assessed in a cytokine release assay after 24-h stimulation of whole blood with a MERS-CoV spike-specific peptide pool. **b** IFN-γ levels in vaccinated individuals (purple, $n = 40$) compared to the placebo group (grey, $n = 4$) at V3M24 ($p = 0.011$). **c** IL-2 levels in vaccinated individuals (purple, $n = 40$) compared to the placebo group (grey, $n = 4$) at V3M24 ($p = 0.0023$). Data were compared using the

Mann–Whitney-U test. The dotted vertical lines show the predefined, cytokine-specific lower limit of quantification (**b**, **c**). **d** Spearman correlation of IFN-γ and IL-2 responses ($n = 44$, $r = 0.87$, $p < 0.0001$). Statistical tests are two-sided. V vaccination, M month, *$p < 0.05$, **$p < 0.01$. Source data are provided as a Source Data file. Created in BioRender. Mayer, L. (2026) https://BioRender.com/oaiwoms.

antibodies, 79% of participants remained seropositive for S1 IgG, and ~60% of participants had a positive T cell response[9]. Nonetheless, a direct comparison between these studies is difficult because of the use of different vaccination schedules and immunogenicity assays.

In our MVA-MERS-S trial, we observed that antibody titers were substantially boosted by V3 to comparable levels between treatments groups, waned more slowly, and remained higher compared to antibody levels observed after V2. Our data suggest, that the V1-V2 interval and vaccine dose do not impact peak antibody titers and antibody persistence as long as V3 is administered. However, the small sample size within each study group makes it challenging to detect significant differences. We previously reported the enrichment of activated, S-specific memory B cells and persistent antibody titers in a subset of seven participants of the MVA-MERS-S phase 1a trial, who received a booster dose (V3) one year after their initial prime-boost vaccination[5].

Recall responses of memory B cells are more efficient at producing long-lived plasma cells, which could explain the observed increased magnitude and durability of the antibody response after V3[21]. This enhanced antibody persistence following a third dose is consistent with findings from COVID-19 vaccination studies, where a third dose translated into higher long-term effectiveness against disease[22–24].

Interestingly, we found that the magnitude of the peak antibody titer at V3M1 strongly correlated with long-term antibody persistence. In line with our findings, Huttner et al. demonstrated that the magnitude of the peak binding antibody response induced by rVSV-EBOV vaccination was the strongest predictor of 5-year antibody persistence[25]. These data suggest that a vaccine's ability to elicit robust peak antibody titers could be a strong indicator of its long-term humoral immunogenicity. Overall, we observed a substantial variation in peak antibody titers between study participants, indicating that

regardless of vaccination regimen, some individuals responded more robustly than others. This variation was not associated with pre-existing or induced humoral immunity against the MVA viral vector. Although human data are limited, several clinical trials have, in line with our results, reported no impact of anti-vector immunity on the induction of insert-specific humoral immunity[26–28]. There was no difference in sex, age or BMI between high and low responders, but the variability in vaccine response could be attributable to other demographic or genetic factors[29], which warrant further investigation.

While for COVID-19, caused by a related betacoronavirus, antibody responses have been suggested as immune correlates of vaccine-induced protection in humans[11,12], the correlates of protection against MERS remain unknown. Humoral immunity, however, is expected to play a critical role, as MERS-CoV-specific antibody responses have been shown to correlate with protection in mice[30,31], to correlate with reduced MERS-CoV viral load in camels[32], and to persist in MERS survivors two years post infection[33]. Importantly, we could show that sera of MVA-MERS-S-vaccinated individuals effectively cross-neutralized MERS-CoV spike variants D510G and I529T, with no reduction in neutralizing titers compared to the wild-type virus. These polymorphisms emerged and spread in humans during the 2015 Korean outbreak[34,35]. Reduced neutralization of these mutants was previously reported in some but not all tested sera of MERS patients[34,36,37] and escape from monoclonal antibodies targeting the affected epitopes was previously shown[38]. Our data suggests, that MVA-MERS-S elicits a broad neutralizing antibody response that remains effective against variants, which is encouraging for MERS vaccine development. Furthermore, we were able to show that MVA-MERS-S generates durable T cell immunity, which is inherently less prone to viral escape and particularly relevant for protection against severe disease[39–41].

Since all study participants had pre-existing immunity to SARS-CoV-2, a comparative analysis between exposed and unexposed participants was not feasible. We, however, observed no correlation between SARS-CoV-2-specific antibody titers and MVA-MERS-S-specific immunity, and previously showed that neutralizing and S1-specific antibody responses were comparable with those seen in the pre-pandemic phase 1a trial[10]. Cross-reactive responses have been reported in humans for SARS-CoV-2 and seasonal coronaviruses, but not for SARS-CoV-2 and MERS-CoV, and their contribution to cross-protection remains unclear[42,43]. If B cell or T cell clones against specific conserved betacoronavirus epitopes are preferentially re-expanded upon heterologous challenge is subject to future investigations.

The value of long-term follow-up studies of vaccine trials, such as the one presented here, lies in their ability to longitudinally characterize the durability of vaccine-induced immunity. Such data might be used in the future to predict the long-term efficacy of a vaccine and inform boosting strategies[44]. We calibrated our ELISA and pseudovirus assays to the WHO international standard, facilitating the comparison of our findings with future studies. Standardized measurements of antibody responses may also be crucial for evaluating the efficacy of novel vaccine candidates via immunobridging[45]. This is especially relevant for pathogens like MERS-CoV, where efficacy trials are not feasible due to the currently restricted number of reported cases[46]. A limitation of this study is the absence of a group that did not receive a booster dose, limiting a direct comparison of antibody persistence between the second and third vaccinations to the 6-month timepoint. Additionally, a longitudinal comparison of the T cell response was not feasible as whole blood samples were not available at the early time-points. As this phase 1 trial was conducted in a country where MERS-CoV does not circulate and a comparator group of MERS survivors was not available, future trials will have to assess whether the observed vaccine immunogenicity is protective.

As MVA-MERS-S requires at least two doses to elicit neutralizing antibodies, this vaccine might be less optimal for emergency vaccination schemes in an acute outbreak setting. However, with the strong boosting effect of a third dose, the long-lasting humoral and cellular immunity, and the excellent safety profile of the MVA platform, MVA-MERS-S could be employed to protect the elderly, healthcare workers, abattoir workers and travelers in MERS-CoV-endemic regions.

In conclusion, we demonstrated the two-year persistence of antigen-specific antibody and T cell responses after three-dose vaccination with an MVA-based vaccine candidate against MERS. These findings further support the clinical development of MVA-MERS-S[4,10].

## Methods

### Study design

A placebo-controlled phase 1b clinical trial was conducted between 2021 and 2022 to assess the safety, immunogenicity, and optimal dosing of intramuscularly administered MVA-MERS-S in healthy adults (NCT04119440). The trial protocol was reviewed and approved by the competent authorities in Germany (Paul-Ehrlich-Institute) and the Netherlands (Central Committee on Research Involving Human Subjects) and by the ethics committees of the Hamburg medical association and Erasmus Medical Center. The study complies with all relevant ethical regulations. Written informed consent was obtained from all participants prior to inclusion in the study. Participants were enrolled at study sites in Rotterdam and Hamburg and randomized into four treatment groups, receiving three vaccinations (V1-V3) of either the low dose ($10^7$ PFU) or the high dose ($10^8$ PFU) of MVA-MERS-S. The prime-boost interval (V1-V2) was either 28 or 56 days, followed by a third vaccination (V3) at day 224. To maintain blinding, a placebo dose was administered on day 56 in the 28-day interval groups and on day 28 in the 56-day interval groups. A placebo-only group receiving four placebo doses was included for comparison. The trial was unblinded according to protocol after the last participant last visit, which refers to the one-month timepoint after third vaccination (=V3M1). Safety (primary outcome) and humoral immunogenicity (secondary outcome) data until V3M1 were recently published[10]. The trial was subsequently extended and participants of the Hamburg study site were re-consented to monitor safety and immunogenicity for two years after V3. The trial extension was reviewed and approved by the German Competent National Authority (Paul-Ehrlich-Institute) and the Ethics Committee of the Hamburg Medical Association (reference number 2020-10180-AMG-ff).

Demographic data of trial participants are reported in Table 1 and Supplementary Table 1. Biological sex was self-reported by participants. At each follow-up visit, the occurrence of targeted illness (COVID-19 and any febrile illness), new serious adverse events, targeted concomitant treatment (COVID-19, Mpox, or MVA vaccination, any other vaccination within four weeks before a visit, as well as immunosuppressive therapy defined as >14 days treatment with immune suppressants or other immune-modifying drugs), and travel to MERS-CoV-endemic regions were documented. These data were reported in electronic case report forms (eCRF, secuTrial database version 5.1.0.20). Serum samples were collected at baseline and at months 6 (M6), 12 (M12), and 24 (M24) following V3 to assess antibody responses. Lithium-heparin whole blood samples were collected at V3M24 to assess the durability of the T cell response.

### MERS-CoV-specific ELISAs

MERS-CoV S1- and full-spike-specific binding antibody responses were measured using in-house enzyme-linked immunosorbent assays (ELISAs)[10]. 96-well plates were coated with with 1 µg/ml MERS-CoV S1 protein (Bio Techne, cat. No. 10737-CV-100) or trimerized, pre-fusion stabilized full spike protein (Keith Chapell, University of Queensland, Australia) at 4 °C overnight. After blocking the plates with 5% powdered milk in Tris-buffer for one hour, serum samples (dilution 1:100) were added and incubated for one hour. Detection was done using HRP-labeled rabbit anti-human IgG (Dako, cat. No. P0214) followed by 3,3′,5,5′-Tetramethylbenzidine (TMB, KPL, cat. No. 507603). The

reaction was stopped after 5 min using 0.5 N sulfuric Acid (Merck, cat. No. 1·09073·1000). The optical density was measured at 450 nm using a microplate reader (Tecan Infinite F200). Results were interpolated using a 5-parameter sigmoidal curve fit according to the MERS-CoV IgG international standard, available from the National Institute for Biological Standards and Control (NIBSC, Hertfordshire, United Kingdom, cat. No. 19/178)[47] and reported in international units (IU)/ml. The positivity cut-offs for the S1 and full-spike ELISA were defined as the mean concentration plus three times the standard error of 77 healthy, pre-pandemic sera.

### MERS-CoV-specific neutralization assays

Neutralization of the wild-type MERS-CoV was assessed using a normalized pseudovirus 50% focus reduction neutralization assay ($nFRNT_{50}$) and a live-virus neutralization test ($VNT_{100}$)[10]. For the $nFRNT_{50}$ assay, serial dilutions of heat-inactivated serum samples in Opti-MEM medium (supplemented with 10% fetal bovine serum, penicillin and streptomycin (P/S) and 10 mM Y-27632 apoptosis inhibitor (MedChemExpress, cat. No. HY-10583)) were mixed with 600 focus-forming units (FFU) of MERS-CoV spike-pseudotyped vesicular stomatitis virus (VSV) with Green Fluorescence Protein (GFP)[48] and incubated for 1 h at 37 °C. The mixture was then added to a confluent monolayer of Calu-3 cells in 96-well plates. After overnight incubation, cells were fixed (4% paraformaldehyde) stained (Hoechst 33342 dye, ThermoScientific, cat. No. 62249), imaged (automated confocal microscope Opera Phenix, Perkin Elmer), and GFP-expressing cells were quantified (image analysis software Harmony version 4.9, Perkin Elmer). Serum was considered to be neutralizing at a 50% reduction of GFP-expressing cells. Neutralizing titers were calculated using 5-parameter logistic regression and results were normalized according to the MERS-CoV IgG international standard, available from the National Institute for Biological Standards and Control (NIBSC, Hertfordshire, United Kingdom, cat. No. 19/178)[47]. The lower limit of detection for the $nFRNT_{50}$ is 16 international units (IU)/ml. Samples without neutralization were set to 16 IU/ml.

For the $VNT_{100}$ assay, serial dilutions of heat-inactivated serum samples were incubated with 100 plaque-forming units (PFU) of MERS-CoV (EMC/2012 isolate) followed by addition of Huh-7 cells (JCRB0403). Neutralization was defined as the absence of cytopathic effect four days post-infection. A human monoclonal antibody (Human anti-MERS spike, m336, Detai Bio-Tech Co., Nanjing, China) was used as a neutralization control. Neutralization titers were calculated as reciprocal geometric mean titers of three replicates. The lower limit of detection for the $VNT_{100}$ is a titer of 8. A titer of 8 is considered positive and samples without neutralization were set to a titer of 4.

### Neutralization of MERS-CoV variants

Cross-neutralization of MERS-CoV variants was assessed by $FRNT_{50}$. Vesicular stomatitis virus-based pseudotypes (VSVpp) expressing mutant MERS-CoV spike proteins harboring the D510G or the I529T polymorphism were generated by transfection of HEK-293T cells (ATCC, CRL-3216) with plasmids carrying the mutant spike sequences and inoculation with VSV*ΔG expressing GFP and firefly luciferase[49]. VSVpp were incubated with serial dilutions of study participants' sera for 1 h at 37 °C and applied onto Calu-3 cell (ATCC, HTB-55) monolayers for 18 h (1:1 mixture; 200 FFU/well). After 4% paraformaldehyde fixation and DAPI staining, infection foci were quantified using the iSpot reader (AID) and normalized to the serum-free control. All samples were tested in duplicate.

### SARS-CoV-2-specific antibody assays

SARS-CoV-2-specific antibody responses were measured using a qualitative anti–nucleocapsid IgG/M/A assay (Elecsys, Roche) with a predefined cut-off index (COI) for positivity of ≥1 and a quantitative anti-trimeric-spike IgG assay (LIAISON, DiaSorin) with a predefined cut-off for positivity of 13 arbitrary (arb.) units /ml. The assays were performed according to the manufacturers' instructions.

### MVA-specific neutralization assay

MVA-specific antibody responses were measured using a recombinant rMVA-GFP virus 50% focus reduction neutralization assay (MVA $FRNT_{50}$)[10]. Serial dilutions of heat-inactivated serum samples in serum-free AdDF-12 medium (supplemented with P/S and HEPES) were incubated with 1500 FFU of rMVA-GFP at 37 °C for one hour and then added to Vero cells (seeded at a density of 30.000 cells/ well the previous day) in 96-well plates. After three days, cells were fixed (4% paraformaldehyde) stained (Hoechst 33342 dye, ThermoScientific Cat. No. 62249), imaged (automated confocal microscope Opera Phenix, Perkin Elmer), and GFP-expressing cells were quantified (image analysis software Harmony version 4.9, Perkin Elmer). A 50% reduction of GFP-expressing cells was considered neutralizing and titers were estimated around the 50% reduction threshold. Non-neutralizing serum samples were given a titre of 10. Non-neutralizing samples were given a titer of 10.

### Cytokine-release T cell assay

Spike-specific T cell responses were measured using a cytokine-release assay. Lithium-heparin whole blood was stimulated with a MERS-CoV spike peptide pool consisting of 15-mers overlapping by 11 amino acids (GenBank: JX869059; JPT Peptide Technologies; 1 µg/ml) for 20–24 h at 37 °C. After stimulation, blood samples were centrifuged for 10 min at $12,000 \times g$ and IFN-γ and IL-2 secretion was measured in duplicate in the supernatant using a microfluidic, multiplex ELISA following the manufacturer's instructions (ELLA, ProteinSimple). Data are shown as concentrations after background subtraction of an unstimulated control for each blood sample. Samples with undetectable cytokine concentrations were set to the lower limit of quantification predefined by the assay standard curve.

### Statistical analysis

Seropositivity was defined as an antibody titer at least fourfold above baseline (pre-vaccination) levels. The Friedman test was used to compare paired samples, whereas unpaired samples were compared using the Mann–Whitney $U$ test or the Kruskal–Wallis test. Statistical tests were two-sided and a $p$-value of ≤0.05 was considered statistically significant. Where applicable, statistical significance was adjusted using Dunn's multiple comparisons test. Associations between data sets were assessed using Spearman correlation or simple linear regression. Figures show measurements of individual study participants. Data analysis was performed using GraphPad Prism version 10.4.2 and Microsoft Excel version 1808. Figures were created using GraphPad Prism version 10.4.2 and BioRender.com.

### Reporting summary

Further information on research design is available in the Nature Portfolio Reporting Summary linked to this article.

## Data availability

The de-identified, individual-level immunogenicity data generated in this study have been deposited in the ZFDM repository database of the University of Hamburg under accession code 17971 and are provided in the Supplementary Information/Source Data file. De-identified, individual-level demographic and clinical data are not publicly available due to data privacy laws. Access for non-commercial purposes can be requested by researchers from the corresponding author (l.mayer@uke.de) and will be granted upon signing of a data transfer agreement. An initial response to data access requests will be given

within four weeks. The original clinical study protocol is available in the supplementary information. Source data are provided with this paper.

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

## Acknowledgements

We would like to thank all volunteers for their dedication and participation in the trial which started and was conducted during the COVID-19 pandemic. We also thank the members of the MVA-MERS-S_DF1 study group for their contributions to this study. A list of members and their affiliations is provided in the supplementary information. This research was supported by funding from the Coalition for Epidemic Preparedness Innovations (CEPI), MMA; the German Centre for Infection Research (DZIF) Thematic Translational Unit (TTU) 01.945 and 01.709, MMA, S.H.; and the Deutsche Forschungsgemeinschaft (DFG, German Research Foundation) –SFB 1648/1 2024 –512741711, M.M.A. We acknowledge financial support from the Open Access Publication Fund of UKE - Universitätsklinikum Hamburg-Eppendorf.

## Author contributions

Writing – original draft: L.M. Writing – review and editing: S.H., A.F., L.M.W., I.G., G.K.G., M.M.A., M.R.t.M. Investigation – safety: H.M.W., I.G., C.S., L.M., M.M.A. Investigation – immunogenicity: A.M., M.R.t.M., G.K.G., J.R., M.L., the MVA-MERS-S_DF1 study group. Data analysis: L.M. Study design & coordination: A.F., M.P.R., C. Dahlke, L.M.W., L.M., B.L.H., S.B., S.H., M.M.A., H.M.W., C.S., C. Drosten, the MVA-MERS-S_DF1 study group. Funding acquisition: S.H., M.M.A.

## Funding

## Competing interests

A.F. became a full-time employee of BioNTech in January, 2024, after completion of her contribution to this study. C. Dahlke was recruited as the translational immunology lead at the Coalition for Epidemic Preparedness Innovations (CEPI) in November, 2022, with responsibilities separate from her role in the present study. B.L.H. is an inventor on a MERS patent with Erasmus MC Rotterdam (No. WO2014045254 A3). S.B. is named as an inventor on a patent for a novel MERS-CoV vaccine (No. EP3045181A1). All other authors declare no competing interests.
