## [Peer Review file · Nature Communications]

Two-year persistence of MERS-CoV-specific antibody and T cell responses after MVA-MERS-S vaccination in healthy adults

Corresponding Author: Dr Leonie Mayer

Version 0:

Reviewer comments:

Reviewer #1

(Remarks to the Author)

The authors evaluated the MVA-MERS-S vaccine candidate in a phase 1b clinical trial to assess its safety and immunogenicity. A two-dose regimen elicited strong spike-specific antibody responses, which were further enhanced by a third booster dose. In a two-year follow-up, antibody titers gradually declined but remained detectable, with levels comparable to those observed after the second dose. Although the precise immune correlates of protection for MERS remain unclear, the durability of the antibody responses supports the potential of MVA-MERS-S as a promising vaccine candidate. The findings also emphasize the importance of booster immunization in maintaining long-term immunity. The study is well-structured and employs appropriate immunological assays. Nonetheless, I have several suggestions to strengthen the scientific implications of the findings, particularly in the context of ongoing beta-coronavirus (bCoV) vaccine development:

The global circulation of SARS-CoV-2 and its variants may influence immune responses to MERS-CoV through potential cross-reactivity among bCoVs. This has been documented in several studies. Notably, a substantial proportion of participants in this cohort had SARS-CoV-2 infection or vaccination during the study period. This presents a valuable opportunity to evaluate cross-reactive immunity. However, the authors did not assess SARS-CoV-2-specific antibody responses. I strongly recommend measuring SARS-CoV-2 binding and neutralizing antibodies using stored sera. Furthermore, a comparative analysis of MERS-CoV antibody titers and neutralization activity between SARS-CoV-2–exposed and –unexposed participants would provide additional insight into bCoV cross-reactivity.

MVA has been used as a viral vector in vaccines against smallpox, mpox, and other infectious diseases. A recognized concern is that pre-existing immunity to the MVA vector may dampen immune responses to the encoded antigen—in this case, the MERS-CoV spike protein. The study also reported variability in antibody responses (high and low responders), which was not fully explained. The authors are encouraged to evaluate anti-MVA antibody levels prior to vaccination to determine whether pre-existing vector immunity correlates with reduced vaccine responsiveness.

As noted in the introduction, other MERS vaccine candidates have been tested in clinical trials, including MVA-MERS-S, ChAdOx1 MERS, and GLS-5300 DNA. A brief comparison of these platforms with respect to antibody durability and the impact of booster doses would help contextualize the current findings and highlight the advantages of MVA-MERS-S.

Previous studies have identified neutralizing antibody escape mutations in MERS-CoV during human-to-human transmission (mBio. 2016 Mar 1;7(2):e00019; Emerg Infect Dis. 2019 Jun;25(6):1161-1168). It would be valuable to investigate whether antibodies elicited by MVA-MERS-S are capable of neutralizing these escape variants. The authors could consider using pseudoviruses expressing spike proteins with these mutations in neutralization assays.

The current study did not evaluate MERS-CoV spike-specific cellular immune responses. If peripheral blood mononuclear cells (PBMCs) were collected from participants, assessing cellular immunity and its kinetics would significantly enhance the immunological profile of this vaccine.

Finally, the authors should clarify how the lower limit of detection or cutoff titers for antibody responses were determined. Please provide detailed criteria in the Methods section.

Reviewer #2

(Remarks to the Author)

Reviewer's comments

MVA-MERS-S, a MERS vaccine candidate, was evaluated in a phase 1b clinical trial for safety, immunogenicity, and optimal dosing in healthy adults. A two-dose regimen generated strong spike-specific antibody responses, which were further boosted by a third dose, with antibodies remaining detectable for at least 24 months. These findings support MVA-MERS-S as a promising vaccine candidate and underscore the role of booster doses in maintaining long-term immunity. The manuscript is generally clear, well-structured, and supported by appropriate data. I just have some major and minor points for the authors.

Major Comments:

Discussion:

- Consider explicitly mentioning potential impact for MERS outbreak preparedness or use as a platform for other coronaviruses.

- A table comparing persistence across MVA-MERS-S, ChAdOx1 MERS, and GLS-5300, if available, would add comparative value.

Figures/Table

-The authors may consider adding a summary figure comparing antibody dynamics between doses 2 and 3.

-The authors may consider adding a demographic comparison of responders vs non-responders at V3M24.

Minor comments:

Title: Consider specifying "in healthy adults" to highlight the population studies.

-For consistency: choose of using either "two-year flow-up" or "24-month flow-up" in your text.

-The authors may also expand on the limitations of the study, including a lack of immunogenicity comparison between natural infection beyond just seropositivity and unknown protective thresholds of MERS.

Reviewer #3

(Remarks to the Author)

this is a well-designed study and had showed persistence of immune response V3M24. Although the sample size was small. The authors did mention some of the limitation but did not comment if COVID vaccine or COVID disease could impact the immune response to the MERS vaccine. these need to be discussed to further enhance the manuscript.

Version 1:

Reviewer comments:

Reviewer #1

(Remarks to the Author)

The authors have provided thorough and appropriate responses to the reviewers' comments, conducted the requested experiments, and substantially expanded the discussion, thereby enhancing the overall quality of the study.

Reviewer #4

(Remarks to the Author)

The authors have properly addressed all the comments from Reviewer 2.

Point-by-point response

Reviewer comments:

Reviewer #1 (Remarks to the Author):

The authors evaluated the MVA-MERS-S vaccine candidate in a phase 1b clinical trial to assess its safety and immunogenicity. A two-dose regimen elicited strong spike-specific antibody responses, which were further enhanced by a third booster dose. In a two-year follow-up, antibody titers gradually declined but remained detectable, with levels comparable to those observed after the second dose. Although the precise immune correlates of protection for MERS remain unclear, the durability of the antibody responses supports the potential of MVA-MERS-S as a promising vaccine candidate. The findings also emphasize the importance of booster immunization in maintaining long-term immunity. The study is well-structured and employs appropriate immunological assays. Nonetheless, I have several suggestions to strengthen the scientific implications of the findings, particularly in the context of ongoing beta-coronavirus (bCoV) vaccine development:

Reply:

We thank the reviewer for the critical reading of our manuscript and the valuable suggestions for improvement. We have included additional data and carefully revised the text and figures to address these concerns.

1. **The global circulation of SARS-CoV-2 and its variants may influence immune responses to MERS-CoV through potential cross-reactivity among bCoVs. This has been documented in several studies. Notably, a substantial proportion of participants in this cohort had SARS-CoV-2 infection or vaccination during the study period. This presents a valuable opportunity to evaluate cross-reactive immunity. However, the authors did not assess SARS-CoV-2-specific antibody responses. I strongly recommend measuring SARS-CoV-2 binding and neutralizing antibodies using stored sera.**

Reply:

We thank the reviewer for this important comment. We now measured SARS-CoV-2 spike-specific binding antibody titers at baseline, to assess the extent of previous SARS-CoV-2 exposure in study participants. Notably, 98% (47/48) of participants were already SARS-CoV-2 seropositive at baseline before MVA-MERS-S vaccination (Fig. 1a, below), suggesting that they had been exposed to SARS-CoV-2 by infection and/ or vaccination already before inclusion into the study. We then analyzed SARS-CoV-2 nucleocapsid-specific binding antibodies at the follow-up timepoints to assess the extent of SARS-CoV-2 infections during the study. Even though only 56% (27/48) of participants had self-reported a SARS-CoV-2 infection, we observed an increase in nucleocapsid antibody titers between the sampling time points in 88% (42/48) of participants (Fig. 1b, below), indicating that SARS-CoV-2 infections were very frequent during the study. We have now added this SARS-CoV-2 antibody data as Supplementary Figure 5 to the manuscript. We have also measured SARS-CoV-2 spike binding antibody responses at the follow-up timepoints and this data can be included in the supplementary material upon request by the reviewer. As spike-binding and neutralizing antibodies show a significant positive correlation, we believe that measuring neutralizing antibodies would not provide additional insights. As shown in Figure 1c-f, baseline SARS-CoV-2-specific antibody responses showed no correlation with MERS-CoV-specific binding and neutralizing antibodies elicited by MVA-MERS-S vaccination at V3M1 and V3M24, suggesting that previously acquired SARS-CoV-2-specific titers did not impact the magnitude and persistence of vaccine-induced antibody responses.

To evaluate if previously induced, cross-reactive responses are preferentially expanded upon heterologous infection or vaccination, would require the identification of cross-reactive B cell epitopes and the isolation and characterization of reactive clones. These analyses are subject of future investigations and beyond the scope of this manuscript. We have now acknowledged this in the discussion. We added the following text to the manuscript:

Methods:

Lines 278-282: “SARS-CoV-2-specific antibody assays

SARS-CoV-2-specific antibody responses were measured using a qualitative anti-nucleocapsid IgG/M/A assay (Elecsys, Roche) with a predefined cut-off index (COI) for positivity of ≥ 1 and a quantitative anti-trimeric-spike IgG assay (LIAISON, DiaSorin) with a predefined cut-off for positivity of 13 arbitrary units (AU)/ml.”

Results:

Lines 116-123: “In the immunogenicity set, 47/48 (98%) participants were SARS-CoV-2 seropositive at baseline before MVA-MERS-S vaccination (Supplementary Fig. 5a) and 42/48 (88%) showed an increase in SARS-CoV-2 nucleocapsid antibodies indicative of a SARS-CoV-2 infection during the 24-month follow-up (Supplementary Fig. 5b). There was no significant correlation between pre-existing SARS-CoV-2 S IgG titers and MERS-CoV S1 IgG at V3M1 ($r = -0.017$, 95% CI: $-0.31-0.28$, $p = 0.91$, Supplementary Fig. 5c) and V3M24 ($r = 0.14$, 95% CI: $-0.17-0.43$, $p = 0.37$, Supplementary Fig. 5d), as well as MERS-CoV neutralizing antibody titers at V3M1 ($r = 0.020$, 95% CI: $-0.27-0.31$, $p = 0.90$, Supplementary Fig. 5e) and V3M24 ($r = 0.11$, 95% CI: $-0.20-0.40$, $p = 0.48$, Supplementary Fig. 5f).”

Discussion:

See reply to comment 2 below.

Figure 1: SARS-CoV-2-specific antibody response. a) Anti-spike IgG at baseline before MVA-MERS-S vaccination. b) Anti-nucleocapsid IgG/M/A at the long-term follow-up timepoints. Correlation between SARS-CoV-2 spike (S)-specific IgG at baseline and MERS-CoV S1 IgG at V3M1 (c) and V3M24 (d), as well as MERS-CoV neutralizing titers at V3M1 (e) and V3M24 (f). COI = cut-off index; AU = arbitrary units, nAb = neutralizing antibody.

2. Furthermore, a comparative analysis of MERS-CoV antibody titers and neutralization activity between SARS-CoV-2-exposed and -unexposed participants would provide additional insight into bCoV cross-reactivity.

Reply:

We agree with the reviewer that a comparative analysis between SARS-CoV-2 exposed and unexposed participants would be of high interest to assess cross-reactivity. However, this analysis is not possible within this clinical trial, as all study participants were SARS-CoV-2-exposed (as outlined above). However, we previously conducted a first-in-human phase 1a trial prior to the COVID-19 pandemic¹. A comparison of this pre-pandemic phase 1a trial with the here-described post-pandemic phase 1b trial provides a unique opportunity to compare the effect of previous SARS-CoV-2 exposure on MERS-CoV-specific immunity. This comparison (Fig. II, below, published by Raadsen et al.²) shows that antibody titers against the full-length MERS-CoV spike (A) were detectable at baseline before MVA-MERS-S vaccination in post-pandemic (phase 1b, pink & green), but not in pre-pandemic samples (phase 1a, orange & blue). This suggests that SARS-CoV-2-specific antibodies cross-react with the full-length MERS-CoV spike protein. This can be explained by the sequence homology of the S2 subunits of the MERS-CoV and SARS-CoV-2 spike proteins. However, no cross-reactivity was seen in the S1-specific MERS-CoV ELISA (B) or the MERS-CoV nFRNT₅₀ neutralization assay (C). As the aim of the here-presented study was to analyze if MVA-MERS-S elicits durable MERS-CoV-specific immunity, we focused the immunogenicity analyses on the MERS-CoV S1 ELISA and the MERS-CoV nFRNT neutralization assay, which are highly specific for MERS-CoV.

We have added the following text to the manuscript:

Discussion:

Lines 208-215: “Since all study participants had pre-existing immunity to SARS-CoV-2, a comparative analysis between exposed and unexposed participants was not feasible. We, however, observed no correlation between SARS-CoV-2-specific antibody titers and MVA-MERS-S immunogenicity, and previously showed that neutralizing and S1-specific antibody responses were comparable with those seen in the pre-pandemic phase 1a trial². Cross-reactive responses have been reported in humans for SARS-CoV-2 and seasonal coronaviruses, but not for SARS-CoV-2 and MERS-CoV, and their contribution to cross-protection remains unclear^{3,4}. If B cell or T cell clones against specific conserved betacoronavirus epitopes are preferentially re-expanded upon heterologous challenge is subject to future investigations. “

Figure II: Comparison of MERS-CoV antibody responses in participants of the pre-pandemic phase 1a trial (orange = low dose; blue = high dose) and the post-pandemic phase 1b trial (pink = low dose, green = high dose) at baseline and after the first and second dose of MVA-MERS-S. A) Binding antibodies against the full-length MERS-CoV spike, B) binding antibodies against the S1 subunit of the MERS-CoV spike, C) MERS-CoV pseudovirus neutralizing antibody titers. Solid horizontal black lines indicate geometric means, error bars indicate 95% confidence intervals. Lower limits of detection are indicated with dotted horizontal lines.

- MVA has been used as a viral vector in vaccines against smallpox, mpox, and other infectious diseases. A recognized concern is that pre-existing immunity to the MVA vector may dampen immune responses to the encoded antigen—in this case, the MERS-CoV spike protein. The study also reported variability in antibody responses (high and low responders), which was not fully explained. The authors are encouraged to evaluate anti-MVA antibody levels prior to vaccination to determine whether pre-existing vector immunity correlates with reduced vaccine responsiveness.**

Reply:

We thank the reviewer for the valuable suggestion to evaluate the effect of preexisting anti-MVA immunity on vaccine immunogenicity. We have now measured MVA-specific antibody titers at baseline prior to first vaccination and prior to the booster vaccination (V3), and have performed a

correlation analysis with peak MERS-CoV S1-specific antibody titers elicited by MVA-MERS-S vaccination. As shown in Figure III below, there is no significant correlation between anti-MVA neutralizing antibodies at baseline prior to first vaccination and vaccine-induced MERS-CoV S1-specific antibodies at the peak V3M1 timepoint (a) and the long-term follow-up timepoint V3M24 (b). Since only three participants had detectable anti-MVA titers at baseline before V1, we extended our analysis to measure anti-MVA titers before V3, to assess whether anti-MVA titers that were induced by V1 and V2 negatively affect responsiveness to V3. Interestingly, there was also no significant correlation between anti-MVA neutralizing antibodies before V3 (= V2M6 timepoint) and vaccine-induced MERS-CoV S1-specific antibodies at the peak V3M1 timepoint (c) and the long-term follow-up timepoint V3M24 (d). We thus do not believe, that anti-MVA immunity negatively affects vaccine responsiveness as measured by humoral immune responses.

Although human data on the impact of anti-vector immunity is limited, several studies have made observations in line with our results. In a trial of an MVA-vectored CMV vaccine, there was no difference in the induction of CMV-specific antibody and T cell responses in participants with pre-existing vaccinia virus immunity due to smallpox vaccination⁵. In another study it was shown that despite the induction of anti-MVA immunity, a series of three homologous MVA-vectored influenza vaccinations could sequentially boost influenza-specific antibody response⁶. An analysis of the ChAdOx1 nCoV-19 phase 1/2 trial showed that ChAdOx1-neutralizing antibody titers induced by the first vaccination did not correlate with spike-specific antibody and T cell responses elicited by the second vaccination⁷.

We have now included the MVA-specific antibody data as Supplementary Figure 6, and added the following text in the manuscript:

Methods:

Lines 283-286: “MVA-specific neutralization assay

MVA-specific antibody responses were measured using a recombinant rMVA-GFP virus 50% focus reduction neutralization assay (MVA FRNT₅₀), as previously described². A 50% reduction of GFP-expressing cells was considered neutralizing. Non-neutralizing samples were given a titer of 10. “

Results:

Lines 124-129: “We observed no significant correlation between pre-existing neutralizing antibody titers against the MVA vector and peak S1 IgG titers at V3M1 ($r = -0.17$, 95% CI: $-0.44-0.13$, $p = 0.25$, Supplementary Fig 6a) or persistent S1 IgG titers at V3M24 ($r = -0.16$, 95% CI: $-0.45-0.15$, $p = 0.29$ Supplementary Fig. 6b). MVA-neutralizing titers elicited by the first two vaccinations did not correlate with peak S1 IgG titers at V3M1 ($r = 0.19$, 95% CI: $-0.11-0.46$, $p = 0.20$, Supplementary Fig. 6c) or persistent S1 IgG titers at V3M24 ($r = 0.13$, 95% CI: $-0.073-0.50$, $p = 0.38$, Supplementary Fig. 6d).”

Discussion:

Lines 186-189: “This variation was not associated with pre-existing or induced humoral immunity against the MVA viral vector. Although human data are limited, several clinical trials have, in line with our results, reported no impact of anti-vector immunity on the induction of insert-specific humoral immunity⁵⁻⁷.”

Figure III: Anti-vector immunity. Correlation of baseline MVA-neutralizing antibody titers at baseline and a) MERS-CoV S1 IgG at V3M1 (a) and at V3M24 (b). Correlation of MVA-neutralizing antibody titers before V3 and MERS-CoV S1 IgG at V3M1 (c) and V3M24 (d).

4. As noted in the introduction, other MERS vaccine candidates have been tested in clinical trials, including MVA-MERS-S, ChAdOx1 MERS, and GLS-5300 DNA. A brief comparison of these platforms with respect to antibody durability and the impact of booster doses would help contextualize the current findings and highlight the advantages of MVA-MERS-S.

Reply:

We thank the reviewer for this valuable comment and agree, that a comparison between the long-term immunogenicity of the three MERS vaccine candidates would be an important addition to the manuscript. To address this point, we have now specified the overall seropositivity rates at V3M24 in the results section to be able to contextualize our findings with the available data of the ChAdOx1 MERS and the GLS-5300 DNA vaccine trials in the discussion section. We have added the following text to the manuscript:

Results:

Lines 83-85: “In total, 60% (24/40) of vaccinated participants remained S1 IgG seropositive at V3M24, compared to 86% at V3M1 (Supplementary Table 6).”

Lines 86-88: “At V3M24, 75% (30/40) and 50% (20/40) of participants maintained detectable pseudovirus and live-virus neutralizing titers, respectively.”

Discussion:

Lines 151-167: “In this study, we demonstrated that three-dose vaccination with the MVA-MERS-S candidate elicits robust immune responses that persist for at least two years, with 75% and 50% of participants maintaining detectable neutralizing antibodies measured by pseudovirus and live-virus assays, respectively, and 60% of participants remaining S1 IgG seropositive... Two other MERS vaccine candidates, ChAdOx1 MERS and GLS-5300 DNA, have demonstrated antigen-specific immunogenicity in humans, but only limited data on long-term immunity are available. Folegatti et al. reported that one year after single-dose ChadOx1 MERS vaccination, 68% of participants maintained full-spike-specific antibody titers above the assay cut-off and the T cell responses remained significantly above baseline⁸. Booster doses of ChAdOx1 MERS have not been tested in clinical trials; however, in a preclinical model, ChadOx1 MERS-elicited responses could be significantly boosted by subsequent vaccination with an MVA-based vaccine candidate⁹. Modjarrad et al. reported that one year after three-dose GLS-5300 DNA vaccination, 3% of participants had detectable live-virus neutralizing antibodies, 79% of participants remained seropositive for S1 IgG, and approximately 60% of participants had a positive T cell response¹⁰. Nonetheless, a direct comparison between these studies is difficult because of the use of different vaccination schedules and immunogenicity assays.”

- 5. Previous studies have identified neutralizing antibody escape mutations in MERS-CoV during human-to-human transmission (mBio. 2016 Mar 1;7(2):e00019; Emerg Infect Dis. 2019 Jun;25(6):1161-1168). It would be valuable to investigate whether antibodies elicited by MVA-MERS-S are capable of neutralizing these escape variants. The authors could consider using pseudoviruses expressing spike proteins with these mutations in neutralization assays.**

Reply:

Since its emergence in 2012, MERS-CoV has been circulating in camels and several mutant strains have been isolated from camels and some also from human isolates¹¹. Especially mutations in the spike protein are of concern as they could lead to escape from serum neutralization. Notably, two nonsynonymous mutations in the receptor binding domain (RBD) of the MERS-CoV spike, D510G and I529T, emerged in the 2015 South Korea outbreak¹². Kim et al. reported, that sera of three patients infected in Korea with the wild-type MERS-CoV showed on average a 3-fold reduction in neutralization of the I529T mutant^{12,13}. Kleine-Weber et al. reported a reduction in neutralization of the D510G and the I529T mutant in sera of 2/3 and 1/3 patients infected in Saudi Arabia, respectively¹⁴. Addetia et al. reported no significant change in neutralization compared to wild-type in ten MERS patients¹⁵. We have now employed a pseudovirus system expressing spike proteins carrying the D510G or the I529T mutation and tested cross-neutralization in ten serum samples of MVA-MERS-S-vaccinated individuals. As shown in Fig. IV below, there was no significant reduction in neutralization of these variants compared to the wild-type.

Figure IV: Neutralization of MERS-CoV variants. a) Schematic of the MERS-CoV spike protein showing the location of the D510G and I529T polymorphisms within the receptor-binding domain (RBD) of the S1 subunit. Neutralizing antibody titers in participant sera against VSVpp harboring the wild-type MERS-CoV spike or MERS-CoV spike variants. Data are shown as individual points and geometric mean titers with 95% confidence interval. Comparison of neutralization against wild-type and variant D510G (b) and I529T (c) in individual samples. Neutralizing titers were compared using Wilcoxon test. WT = wild-type.

We have now included this additional data as Figure 5 in the manuscript and have added the following text to the manuscript:

Methods:

Lines 270-277: “Neutralization of MERS-CoV variants

Cross-neutralization of MERS-CoV variants was assessed by FRNT₅₀. Vesicular stomatitis virus-based pseudotypes (VSVpp) expressing mutant MERS-CoV spike proteins harboring the D510G or the I529T polymorphism were generated as previously described¹⁶. VSVpp were incubated with serial dilutions of study participants’ sera (1h, 37°C) and applied onto Calu-3 cell monolayers (18h, 1:1 mixture; 200 focus-forming units (FFU)/well). After 4% paraformaldehyde fixation and DAPI staining, infection foci were quantified using the iSpot reader (AID) and normalized to the serum-free control. All samples were tested in duplicate.”

Results:

Lines 130-137: “Cross-neutralization of MERS-CoV spike variants

To assess whether MVA-MERS-S elicits cross-neutralization against clinically relevant MERS-CoV variants, we generated VSVpp harboring spike mutations D510G or I529T, which emerged in the 2015 outbreak in South Korea, and measured neutralization in ten serum samples. As shown in Figure 5a, GMTs against wild-type, mutant D510G and mutant I529T spike were 830 (95% CI: 298-2312), 1091 (95% CI: 355-3355), and 1074 (95% CI: 381-3028), respectively. Compared to the wild-type, there was no significant reduction in neutralization of mutant D510G (Fig. 5b) or mutant I529T (Fig. 5c), indicating, that MVA-MERS-S elicits antibodies that can cross-neutralize these variants. “

Discussion:

Lines 198-207: “Importantly, we could show that sera of MVA-MERS-S-vaccinated individuals effectively cross-neutralized MERS-CoV spike variants D510G and I529T, with no reduction in neutralizing titers compared to the wild-type virus. These polymorphisms emerged and spread in humans during the 2015 Korean outbreak^{13,17}. Reduced neutralization of these mutants was previously reported in some but not all tested sera of MERS patients^{15,17,18} and escape from monoclonal antibodies targeting the affected epitopes was previously shown¹⁹. Our data suggests, that MVA-MERS-S elicits a broad neutralizing antibody response that remains effective against variants, which is encouraging for MERS vaccine development. Furthermore, we were able to show that MVA-MERS-S generates durable T cell immunity, which is inherently less prone to viral escape and particularly relevant for protection against severe disease²⁰⁻²².”

- 6. The current study did not evaluate MERS-CoV spike-specific cellular immune responses. If peripheral blood mononuclear cells (PBMCs) were collected from participants, assessing cellular immunity and its kinetics would significantly enhance the immunological profile of this vaccine.**

Reply:

We thank the reviewer for this suggestion, and agree, that accessing cellular immunity would enhance the immunological profile of this vaccine.

[Redacted]

We therefore developed a more sensitive whole-blood based cytokine release assay, in which the cytokine response of antigen-specific T cells is measured by ELISA in the supernatant of whole blood re-stimulated with the MERS-CoV-spike peptide pool. As shown in the Figure VI below, IFN- γ and IL-2 responses were significantly higher at the V3M24 timepoint in the MVA-MERS-S-vaccinated group compared to the placebo group, indicating that MERS-CoV-spike-specific T cell responses remain detectable for at least two years after vaccination. We have now included this data as Figure 6 in the manuscript and have pointed out in the limitations that measuring two-year kinetics of the T cell response was not feasible.

Figure VI: Persistence of MERS-CoV-specific T cell response. a) The T cell responses was assessed in a cytokine release assay after 24-hour stimulation of whole blood with a MERS-CoV spike-specific peptide pool. IFN- γ (b) and IL-2 (c) levels in vaccinated individuals (purple) and the placebo group (grey) at V3M24 compared using the Mann-Whitney U test. The dotted vertical lines show the predefined, cytokine-specific lower limit of quantification. c) Spearman's correlation of IFN- γ and IL-2 responses. V = vaccination, M = month, IU = international units.

We have added the following text to the manuscript:

Methods:

Lines 287-295: "Cytokine-release T cell assay"

Spike-specific T cell responses were measured using a cytokine-release assay. Lithium-heparin whole blood was stimulated with a MERS-CoV spike peptide pool consisting of 15-mers overlapping by 11 amino acids (GenBank: JX869059; JPT Peptide Technologies; 1 μ g/ml) for 20-24 h at 37°C. After stimulation, blood samples were centrifuged for 10 minutes at 12000 x g and IFN- γ and IL-2 secretion was measured in duplicate in the supernatant using a microfluidic, multiplex ELISA following the manufacturer's instructions (ELLA, ProteinSimple). Data are shown as concentrations after background subtraction of an unstimulated control for each blood sample. Samples with undetectable cytokine concentrations were set to the lower limit of quantification predefined by the assay standard curve."

Results:

Lines 138-147: "Persistence of the MERS-CoV-specific T cell response"

Next, we analyzed if vaccine-induced T cell responses persist until two years after vaccination by restimulating whole blood samples at the V3M24 timepoint with an overlapping spike peptide pool

and measuring cytokine release by ELISA (Fig. 6a). As shown in Figure 6b, compared to the median IFN- γ response of the placebo group (6.3 pg/ml, 1.1-10.4), the median IFN- γ response of the MVA-MERS-S-vaccinated group was significantly higher (31.5 pg/ml, 95% CI: 13.5-55.2, $p = 0.0106$). Similarly, compared to the median IL-2 levels of the placebo group (4.3 pg/ml, 95% CI: 2.2-10.0), the median IL-2 response of the MVA-MERS-S-vaccinated group was significantly higher (35.1 pg/ml, 95% CI: 24.3-49.0, $p = 0.0023$, Fig. 6c). There was a significant positive correlation between the IFN- γ and IL-2 response ($r = 0.87$, 95% CI: 0.77-0.93, $p < 0.0001$, Fig. 6c).“

Discussion:

Lines 155-156: “Furthermore, we could show that vaccine-induced T cells secreting IFN- γ and IL-2, indicative of a Th1-biased response, remain detectable two years.

Lines 205-207: “Furthermore, we could show that MVA-MERS-S generates durable T cell immunity, which is inherently less prone to viral escape and particularly relevant for protection against severe disease²⁰⁻²².“

Lines 225-226: “Additionally, a longitudinal comparison of the T cell response was not feasible as whole blood samples were not available at the early timepoints.”

- 7. Finally, the authors should clarify how the lower limit of detection or cutoff titers for antibody responses were determined. Please provide detailed criteria in the Methods section.**

Reply:

We thank the reviewer for this comment and have now clarified the lower limit of detection and cut-offs for the antibody and T cell assays in the methods section as follows:

Lines 261-263: “The positivity cut-offs for the S1 and full-spike ELISA were defined as the mean concentration plus three times the standard error of 77 healthy, pre-pandemic sera.”

Lines 265-268: “The lower limit of detection for the nFRNT₅₀ is 16 international units (IU)/mL. Non-neutralizing serum samples were set to 16 IU/mL. The lower limit of detection for the VNT₁₀₀ is a titer of 8. A titer of 8 is considered positive and samples without neutralization were set to a titer of 4.”

Lines 279-282: “SARS-CoV-2-specific antibody responses were measured using a qualitative anti-nucleocapsid IgG/M/A assay (Elecsys, Roche) with a predefined cut-off index (COI) for positivity of ≥ 1 and a quantitative anti-trimeric-spike IgG assay (LIAISON, DiaSorin) with a predefined cut-off for positivity of 13 arbitrary units (AU)/ml.”

Reviewer #2 (Remarks to the Author):

MVA-MERS-S, a MERS vaccine candidate, was evaluated in a phase 1b clinical trial for safety, immunogenicity, and optimal dosing in healthy adults. A two-dose regimen generated strong spike-specific antibody responses, which were further boosted by a third dose, with antibodies remaining detectable for at least 24 months. These findings support MVA-MERS-S as a promising vaccine candidate and underscore the role of booster doses in maintaining long-term immunity. The manuscript is generally clear, well-structured, and supported by appropriate data. I just have some major and minor points for the authors.

Reply:

We thank the reviewer for critically evaluating our manuscript and for the positive feedback. We have now addressed the major and minor comments as outlined below, including additional comparisons of V2-V3 antibody dynamics and demographics of responders vs. non-responders.

Major Comments:

Discussion:

- 1. Consider explicitly mentioning potential impact for MERS outbreak preparedness or use as a platform for other coronaviruses.**

Reply:

We thank the reviewer for this important suggestion. We have now elaborated on the advantages, limitations, and potential future use of MVA-MERS-S in the discussion and added the following text to the manuscript.

Discussion:

Lines 230-234: “As MVA-MERS-S requires at least two doses to elicit neutralizing antibodies, this vaccine might be less optimal for emergency vaccination schemes in an acute outbreak setting. However, with the strong boosting effect of a third dose, the long-lasting humoral and cellular immunity, and the excellent safety profile of the MVA platform, MVA-MERS-S could be employed to protect the elderly, healthcare workers, abattoir workers and travelers in MERS-CoV-endemic regions. “

- 2. A table comparing persistence across MVA-MERS-S, ChAdOx1 MERS, and GLS-5300, if available, would add comparative value.**

Reply:

We fully agree with the reviewer, that a comparison of antibody persistence across the three vaccine candidates is essential to contextualize our findings. Limited long-term follow-up data until one year after vaccination is reported by Folegatti et al.⁸ and Modjarrad et al.¹⁰ We have now included these data in the discussion lines 150-167, as outlined in the reply to reviewer #1, comment 4 above. We would like to refrain from providing a comparative table, as a direct comparison of the data would mislead the readers, as different vaccination schedules, antibody assays, cut-offs and seropositivity criteria were used in these studies.

Figures/Table:

- 3. The authors may consider adding a summary figure comparing antibody dynamics between doses 2 and 3.**

Reply:

We thank the reviewer for this suggestion; however, we are not entirely sure what kind of figure is requested. Figure 3 shows the MERS-CoV S1-specific antibody dynamics after V2 and V3, allowing for a visual comparison and summarizes the data. The corresponding geometric mean titers are provided in Supplementary Table 5 for further comparison of the underlying data. We kindly ask the reviewer to respecify the requested comparison, in case it is not covered by the provided figures and tables.

- 4. The authors may consider adding a demographic comparison of responders vs. non-responders at V3M24.**

Reply:

We thank the reviewer for this thoughtful suggestion. We have now added a demographic comparison of low and high responders (shown in Supplementary Figure S5) as Supplementary Table 7. We observe no significant difference in sex, age and BMI, as shown below.

Supplementary Table 7: Demographics of immunogenicity set by vaccine response. Summary of the demographic characteristics of participants of the long-term follow-up that met the modified intention-to-treat definition, stratified by the S1 IgG response at V3M1 into low and high responders. Data were collected at the screening visit before V1. V = vaccination, M = month, SD = standard deviation, PFU = plaque-forming units.

	Low responders (n = 22)	High responders (n = 21)
Sex		
Male – n (%)	11 (50%)	11 (52%)
Female – n (%)	11 (50%)	10 (48%)
Age – years		
mean (SD)	34 (\pm 11.2)	35 (\pm 11.2)
Body-mass index		
mean (SD)	23 (\pm 2.6)	23 (\pm 2.6)

We added the following text in the discussion to address this comment:

Results:

Line 110: “A demographic comparison of high and low responders is provided in Supplementary Table 7.”

Discussion:

Lines 189-192: “There was no difference in sex, age or BMI between high and low responders, but the variability in vaccine response could be attributable to other demographic or genetic factors²³, which warrant further investigation.”

Minor comments:

5. **Title: Consider specifying “in healthy adults” to highlight the population studies.**

Reply:

We have now changed the title to highlight the study population as follows:

“Two-year persistence of MERS-CoV-specific antibody and T cell responses after MVA-MERS-S vaccination in ~~humans~~ healthy adults”

6. **For consistency: choose of using either “two-year flow-up” or “24-month flow-up” in your text.**

Reply:

We thank the reviewer for this comment and apologize for the inconsistency. We have now chosen the term “24-month follow-up” throughout the manuscript.

7. **The authors may also expand on the limitations of the study, including a lack of immunogenicity comparison between natural infection beyond just seropositivity and unknown protective thresholds of MERS.**

Reply:

We agree with the reviewer, that the lack of a correlate of protection is a major hurdle in vaccine development, not only against MERS, but against many emerging pathogens. We have now expanded on the limitations of this study, including the lack of a comparator group of MERS survivors, unknown protective antibody thresholds, and overall lack of confirmed correlates of protection for MERS. However, we would also like to highlight, that phase 1 trials are by design not able to answer whether a vaccine confers protection.

We have included the following text in the discussion:

Lines 194-198: “...the correlates of protection against MERS remain unknown. Humoral immunity, however, is expected to play a critical role, as MERS-CoV-specific antibody responses have been shown to correlate with protection in mice^{24,25}, to correlate with reduced MERS-CoV viral load in camels²⁶, and to persist in MERS survivors two years post infection²⁷.”

Lines 226-229: “As this phase 1 trial was conducted in a country where MERS-CoV does not circulate and a comparator group of MERS survivors was not available, future trials will have to assess whether the observed vaccine immunogenicity is protective.”

Reviewer #3 (Remarks to the Author):

This is a well-designed study and had showed persistence of immune response V3M24. Although the sample size was small. The authors did mention some of the limitation but did not comment if COVID

vaccine or COVID disease could impact the immune response to the MERS vaccine. these need to be discussed to further enhance the manuscript.

Reply:

We thank the reviewer for the critical review of our manuscript and the overall positive feedback. As SARS-CoV-2 exposure before and during the study period was widespread within our study cohort, a direct comparison between exposed and unexposed individuals was not feasible. We have now clarified this in the manuscript, with the addition of SARS-CoV-2 serology data, as discussed in detail in the reply to reviewer #2, comment 1. We have also referenced a comparison of antibody responses with our previous phase 1a trial, which was conducted before the COVID-19 pandemic in the reply to reviewer #1, comment 2. This comparison suggests, that MVA-MERS-S-specific immunogenicity is not significantly impacted by SARS-CoV-2 exposure through infection and/or vaccination. This is likely attributed to the antigenic distance between the MERS-CoV and the SARS-CoV-2 spike proteins. If cross-reactive responses could provide (partial) cross-protection against betacoronaviruses is currently highly debated and requires further analyses that are out of the scope of this manuscript. We have now acknowledged this knowledge gap in the discussion.

We have included the following text to the discussion:

Lines 207-216: “Since all study participants had pre-existing immunity to SARS-CoV-2, a comparative analysis between exposed and unexposed participants was not feasible. We, however, observed no correlation between SARS-CoV-2-specific antibody titers and MVA-MERS-S-specific immunity, and previously showed that neutralizing and S1-specific antibody responses were comparable with those seen in the pre-pandemic phase 1a trial²⁸. Cross-reactive responses have been reported in humans for SARS-CoV-2 and seasonal coronaviruses, but not for SARS-CoV-2 and MERS-CoV, and their contribution to cross-protection remains unclear^{3,4}. If B cell or T cell clones against specific conserved betacoronavirus epitopes are preferentially re-expanded upon heterologous challenge is subject to future investigations.”

References

1. Koch, T. *et al.* Safety and immunogenicity of a modified vaccinia virus Ankara vector vaccine candidate for Middle East respiratory syndrome: an open-label, phase 1 trial. *Lancet Infect. Dis.* **20**, 827–838 (2020).
2. Raadsen, M. P. *et al.* Safety, immunogenicity, and optimal dosing of a modified vaccinia Ankara-based vaccine against MERS-CoV in healthy adults: a phase 1b, double-blind, randomised placebo-controlled clinical trial. *Lancet Infect. Dis.* **25**, 231–242 (2025).
3. Murray, S. M. *et al.* The impact of pre-existing cross-reactive immunity on SARS-CoV-2 infection and vaccine responses. *Nat. Rev. Immunol.* **23**, 304–316 (2023).
4. Neto, T. A. P. *et al.* Highly conserved Betacoronavirus sequences are broadly recognized by human T cells. *Cell* S0092-8674(25)00804–9 (2025) doi:10.1016/j.cell.2025.07.015.
5. La Rosa, C. *et al.* MVA vaccine encoding CMV antigens safely induces durable expansion of CMV-specific T cells in healthy adults. *Blood* **129**, 114–125 (2017).
6. Kreijtz, J. H. C. M. *et al.* Safety and immunogenicity of a modified-vaccinia-virus-Ankara-based influenza A H5N1 vaccine: a randomised, double-blind phase 1/2a clinical trial. *Lancet Infect. Dis.* **14**, 1196–1207 (2014).
7. Barrett, J. R. *et al.* Phase 1/2 trial of SARS-CoV-2 vaccine ChAdOx1 nCoV-19 with a booster dose induces multifunctional antibody responses. *Nat. Med.* **27**, 279–288 (2021).

8. Folegatti, P. M. *et al.* Safety and immunogenicity of a candidate Middle East respiratory syndrome coronavirus viral-vectored vaccine: a dose-escalation, open-label, non-randomised, uncontrolled, phase 1 trial. *Lancet Infect. Dis.* **20**, 816–826 (2020).
9. Alharbi, N. K. *et al.* ChAdOx1 and MVA based vaccine candidates against MERS-CoV elicit neutralising antibodies and cellular immune responses in mice. *Vaccine* **35**, 3780–3788 (2017).
10. Modjarrad, K. *et al.* Safety and immunogenicity of an anti-Middle East respiratory syndrome coronavirus DNA vaccine: a phase 1, open-label, single-arm, dose-escalation trial. *Lancet Infect. Dis.* **19**, 1013–1022 (2019).
11. AlBalwi, M. A. *et al.* Evolving sequence mutations in the Middle East Respiratory Syndrome Coronavirus (MERS-CoV). *J. Infect. Public Health* **13**, 1544–1550 (2020).
12. Kim, Y. *et al.* Spread of Mutant Middle East Respiratory Syndrome Coronavirus with Reduced Affinity to Human CD26 during the South Korean Outbreak. *mBio* **7**, e00019-16 (2016).
13. Kim, Y.-S. *et al.* Sequential Emergence and Wide Spread of Neutralization Escape Middle East Respiratory Syndrome Coronavirus Mutants, South Korea, 2015. *Emerg. Infect. Dis.* **25**, 1161–1168 (2019).
14. Kleine-Weber, H. *et al.* Mutations in the Spike Protein of Middle East Respiratory Syndrome Coronavirus Transmitted in Korea Increase Resistance to Antibody-Mediated Neutralization. *J. Virol.* **93**, e01381-18 (2019).
15. Addetia, A. *et al.* Mapping immunodominant sites on the MERS-CoV spike glycoprotein targeted by infection-elicited antibodies in humans. *Cell Rep.* **43**, (2024).
16. Schroeder, S. *et al.* Functional comparison of MERS-coronavirus lineages reveals increased replicative fitness of the recombinant lineage 5. *Nat. Commun.* **12**, 5324 (2021).
17. Kim, Y.-S. *et al.* Sequential Emergence and Wide Spread of Neutralization Escape Middle East Respiratory Syndrome Coronavirus Mutants, South Korea, 2015. *Emerg. Infect. Dis.* **25**, 1161–1168 (2019).
18. Kleine-Weber, H. *et al.* Mutations in the Spike Protein of Middle East Respiratory Syndrome Coronavirus Transmitted in Korea Increase Resistance to Antibody-Mediated Neutralization. *J. Virol.* **93**, e01381-18 (2019).
19. Kleine-Weber, H. *et al.* Mutations in the Spike Protein of Middle East Respiratory Syndrome Coronavirus Transmitted in Korea Increase Resistance to Antibody-Mediated Neutralization. *J. Virol.* **93**, 10.1128/jvi.01381-18 (2019).
20. Pereira Neto, T. A. *et al.* Highly conserved Betacoronavirus sequences are broadly recognized by human T cells. *Cell* S0092-8674(25)00804–9 (2025) doi:10.1016/j.cell.2025.07.015.
21. Sette, A., Sidney, J. & Crotty, S. T Cell Responses to SARS-CoV-2. *Annu. Rev. Immunol.* **41**, 343–373 (2023).
22. Moss, P. The T cell immune response against SARS-CoV-2. *Nat. Immunol.* **23**, 186–193 (2022).
23. Pedroza-Pacheco, I. & McMichael, A. J. Immune signature atlas of vaccines: learning from the good responders. *Nat. Immunol.* **23**, 1654–1656 (2022).
24. Tai, W. *et al.* MERS-CoV RBD-mRNA vaccine induces potent and broadly neutralizing antibodies with protection against MERS-CoV infection. *Virus Res.* **334**, 199156 (2023).
25. Zhao, J. *et al.* Passive Immunotherapy with Dromedary Immune Serum in an Experimental Animal Model for Middle East Respiratory Syndrome Coronavirus Infection. *J. Virol.* **89**, 6117–6120 (2015).
26. Tolah, A. M. *et al.* Cross-sectional prevalence study of MERS-CoV in local and imported dromedary camels in Saudi Arabia, 2016-2018. *PloS One* **15**, e0232790 (2020).
27. Cheon, S. *et al.* Longevity of seropositivity and neutralizing antibodies in recovered MERS patients: a 5-year follow-up study. *Clin. Microbiol. Infect.* **28**, 292–296 (2022).
28. Raadsen, M. P. *et al.* Safety, immunogenicity, and optimal dosing of a modified vaccinia Ankara-based vaccine against MERS-CoV in healthy adults: a phase 1b, double-blind, randomised placebo-controlled clinical trial. *Lancet Infect. Dis.* **0**, (2024).